# A Theory about a Hidden Evander-Size Impact and the Renewal of the Intermediate Cratered Terrain on Dione †

Balázs Bradák [1,*], Mayuko Nishikawa [1] and Christopher Gomez [1,2]

1 Faculty of Oceanology, Kobe University, 5-1-1 Fukaeminami-machi, Higashinada-ku, Kobe 658-0022, Japan
2 Faculty of Geography, Universitas Gadjah Mada, Yogyakarta 55281, Indonesia
* Correspondence: bradak.b@port.kobe-u.ac.jp
† This paper is an extended version from the proceeding paper: Bradák, B.; Nishikawa, M.; Gomez, C. Introduction to a "Radical" Working Hypothesis about a Hemisphere-Scale Impact on Dione (Saturn). In Proceedings of the 2nd Electronic Conference on Universe, Online, 16 February–2 March 2023.

**Abstract:** The study introduces a theory about an Evander-size impact on the surface of Dione. Our study suspects a relatively low-velocity ($\leq 5$ km/s) collision between a ca. 50–80 km diameter object and Dione, which might have resulted in the resurfacing of one of the satellite's intermediate cratered terrains in various ways, such as surface planing by "plowing" by ricocheting ejectiles, ejecta blanket covering, partial melting, and impact-triggered diapir formation associated with cryotectonism and effusive cryo-slurry outflows. Modeling the parameters of an impact of such a size and mapping the potential secondary crater distribution in the target location may function as the first test of plausibility to reveal the location of such a collision, which may be hidden by younger impact marks formed during, e.g., the Antenor, Dido, Romulus, and Remus collision events. The source of the impactor might have been Saturn-specific planetocentric debris, a unique impactor population suspected in the Saturnian system. Other possible candidates are asteroid(s) appearing during the outer Solar System's heavy bombardment period, or a collision, which might have happened during the "giant impact phase" in the early Saturnian system.

**Keywords:** Dione; Saturn; icy satellite; giant impact; Intermediate Cratered Terrain; resurfacing





## 1. Introduction

The discovery of an increasing number of icy satellites orbiting the gas and ice giants of the Solar system with potential subsurface oceans, which may harbor life and provide an environment for biological evolution, has been triggering the intensification of studies targeting those moons [1]. From the sense of a putative living and evolving ecosystem under the "protective" shell of an icy satellite, the oxygenation of their habitat, i.e., the subsurface ocean, is essential. Oxygenation may happen during resurfacing processes, a term covering various types of interaction between the icy surface and the underlying layers via, e.g., asteroid impact-triggered processes, and downward material transportation, supported by cryotectonic processes [2].

To understand the influence and how a resurfacing process, such as a basin-scale asteroid impact, may work, it is essential to describe the generic structure of an icy satellite briefly [1,3,4]. Without going into detail, icy satellites most likely formed in a very similar way to that in which the planets were formed around the evolving Sun: giant planets developed disks of materials which later evolved into the satellite systems of the planets (see [5] and references therein). The first building blocks of Saturn's satellites were probably C- and water-rich chondritic planetesimals. In addition, in the solar-nebula instabilities, appearing in the environment surrounding the giant planets, water-rich comets might also accrete [6]. Icy satellites locating beyond the frost (or snow) line, a boundary beyond which volatiles appear in a condensed, solid state, the key building components of a (subsurface) liquid ocean are frozen [7]. Ocean formation requires a heat source, which in the case of

icy satellites may appear as radioactive decay, tidal heating, or energy associated with the satellite accretion and the early luminosity of the parent body [1]. Along with the balance between internal heating (mostly radioactive decay and tidal heating) and heat removal (convective and conductive ice shell, see below), the maintenance of the subsurface ocean depends on the freezing point of the liquid, i.e., the chemical composition of the subsurface ocean [1,3,4,8]. As a result of those factors, depending on the size of the icy satellite, two scenarios may appear regarding the differentiation of their structure [1,4,8]. Smaller satellites (e.g., Enceladus) develop a liquid ocean horizon under their icy crust, which is in direct contact with the silicate surface of the silicate/iron-rich core innermost part. In contrast, the liquid ocean horizon in larger icy-rich planetary bodies (e.g., Ganymede) is not in direct contact with the innermost silicate-rich sphere; the so-called high-pressure-phase ice horizon is intercalated between the liquid ocean and the innermost, silicate mantle/iron core sections [1]. The outermost solid ice-crust may be differentiated into two characteristic units as well, namely the lower isoviscous convective layer (also referred to as the "ductile convecting ice layer," or, in short, "convective ice") and the upper conductive lid (or simply "conductive ice") [1,3,4,8]. Suppose such differentiation of the icy crust appears. In that case, the so-called "thick-ice model" applies to the satellite, and the crust of icy moons, with the lack of differentiation and the formation of convecting ice, along with the sole development of a conductive ice layer, and is referred to as the "thin-ice model" [8].

Although it has been suspected for a long time, finding evidence about the existence of a subsurface ocean under Dione's icy crust is still an ongoing goal of planetary scientists [9–11]. One of the breakthroughs in the research was the study by Beuthe et al. [10], which, based on the analysis of Dione's gravity-shape data, concluded that a 65 km thick global subsurface ocean is hiding under the approximately 100 km thick crust of the satellite. The calculations of Beuthe et al.'s study [10] were verified by Zannoni et al. [11], supporting the appearance of a liquid subsurface ocean under the icy crust of Dione. Despite its relation (2:1 mean-motion resonance) with her famous brother, Enceladus, who might provide direct evidence of the subsurface ocean by showing its water plumes [12,13], the main chapters of Dione's planetary/geological evolution are poorly understood compared to other icy satellites (e.g., Europa, a moon of Jupiter).

The early geological mapping of the satellite's surface laid the foundation for further studies by identifying the main morphological features of the icy surface and providing a basic chronological framework for their evolution [14,15]. Nearly three decades passed following the first chronological attempt, when new results appeared regarding the evolution of the icy satellite [16], followed by another milestone in Dione's research, especially from the point of studies targeting surface renewal mechanisms on the icy crust of the satellite [15].

In addition, during the chronological classification of various terrains, the study of Kirchoff and Schenk [17] recognized that the so-called intermediate cratered terrains (ITCs) might have been resurfaced at some point during their planetary history. Chronologically, ITCs might have formed between (or following) the formation of the two oldest terrains, the so-called dense cratered terrain (DCT 1) and DCT 2 (both formed approximately 4.1 Ga) [17]. The formation of those terrains, as well as the ICTs, coincide with the expected outer Solar System heavy bombardment period (independently from its spike-like or steady decline nature) (model age of ICT: 3.5 +1.0/−2.6 Ga) [17]. Although the nature of the resurfacing process is still unknown, it might have been able to erase all the previous craters. Some theories have been built to explain such a process, including burial by material "snowing" down from the ring [18,19], thermal activity from tidal dissipation [20–23], and the subsequent formation of larger (D ≥ 50 km) craters and their ejecta blanket [17].

We propose a working hypothesis that suggests a collision with an Evander-size impactor (or bigger) as an alternative cause and trigger of the surface renewal processes. Our research aims to support or deny such a hypothesis by studying the crater distribution patterns on Dione (focusing on one of the ITCs), which may indicate the appearance of secondary crater formation related to the putative impact.

## 2. Materials and Methods

The challenge in identifying a potentially bigger impactor and missing impact crater (the putative cause of surface renewal) on the highly cratered, older ITC surface was to separate the primary and secondary impactors. To solve such a problem, the following hypothesis was built.

During the geological history of Dione, it was exposed to continuous asteroid bombardment, which resulted in a high abundance of craters in certain regions. It can be assumed that the crater distribution pattern related to those impacts is random. During such bombardments, extreme impact events may have happened with irregular-sized impactors resulting in the formation of secondary impact craters. Such secondary craters are formed by the ejecta excavated from the larger crater and located around the original impact center, forming various patterns such as radial crater chains, double and multiple craters, and in the case of an oblique impact, a "path" of craters and surface marks formed by the ricocheting debris. Along such patterns, a theory was built about forming a concentric secondary crater pattern, considering the impact excavation phase and postimpact surface evolution processes. During the excavation phase, various sized debris is thrown out from the crater. Small-sized material, most likely ejected into space, settles into a specific orbit, and/or falls back to the satellite, creating an ejecta blanket. The re-impacting larger debris may create secondary craters right after the primary impact, allocated at various distances from the original impact crater. Regarding the physical characteristics of the impact and the ejectiles, the debris will be deposited (re-impact) concentrically around the impact crater (not necessarily in a complete circle form, depending on the impact angle). For example, an asteroid impact with a given impact energy excavates materials of various sizes and masses, which, given the impact energy, are thrown at various distances around the original impact crater depending on the characteristics of the impact and their physical parameters. In this way, such a process may result in some level of sorting in crater size around the primary crater and the formation of concentric regions around the original impact crater (characterized by the dominance of specific sizes of secondary craters). Phenomena indicating such sorting can be observed, e.g., at Dione's Pantagias Catenae (a region northward of the Evander basin), there is an area with frequent linear crater chain appearance. Observable gradual decrease (increase) in crater size can be observed within the members of such crater chains, most likely related to sorting caused by the influence of various impact and ejectile characteristics. In addition, postimpact processes, such as surface renewal by cryotectonic and cryovolcanic activity and surface relaxation, may result in the preservation of a specific size of craters and the disappearance of others, strengthening the formation of such concentric (ring or fragment-of-a-ring) secondary crater patterns. The concentrically located secondary craters are "added" to the (in general) randomly allocated craters, resulting in an increasing abundance of craters in certain regions, following a concentric (a crater ring or a section of a crater ring) pattern instead of the random distribution. Please note that, in the case of ICT, it may work oppositely. The putative giant impact and its effect possibly erased the earlier craters during the ICT's surface renewal event [17], creating its secondary crater population. The giant impact is followed by further asteroid bombardment adding a random distribution component to the concentric secondary crater pattern.

The mapping of impact craters and the analysis of the distribution of various crater classes may reveal such non-random crater distribution, a possible indicator of a putative impact crater caused by the primary collision.

The following research workflow was built to verify or deny the existence of a most likely hemisphere-to-global-scale impact with no identifiable surface marks.

- Step 1. The idea of using secondary crater rings as the marker of the location of primary impactors. The formation of secondary craters is one of the main topics of the research community involved in impacts and crater formation [24]. It is not limited, e.g., to terrestrial planets and the Moon, but also discussed in the case of icy satellites [25]. In the case of this study, we intend to use secondary crater formation and its allocation (e.g., concentric crater allocation [a crater ring or a section of a crater ring], created

by the ejectiles of the primary impact; Section 2.1.1) as an indicator of the primary impact location.

- Step 2. Modeling the size of impactors in the case of observed crater size (D) $\geq$ 100 km (Section 2.1). The ejectiles and secondary impactors by impacts creating smaller primary craters than the chosen observed crater size may be too small to be recognizable in the frequently cratered surface of the studied location (even if their postimpact fate brings them back to the surface) [26,27].
- Step 3. Determination of the secondary crater formation scenarios (Section 2.1.1). Various impact angles and impact speeds are considered while evaluating the results of secondary crater formation.
- Step 4. Determination of putative secondary-crater size classes. Certain crater allocation patterns of craters belonging to the putative secondary crater classes may indicate various size impactors (e.g., characteristic allocation pattern of craters, fall in crater size of 4–6 km, may indicate an impactor 30–40 km in size).
- Step 5. Mapping of craters in the target location (Section 2.2).
- Step 6. Analysis of the distribution of craters belonging to various putative secondary crater classes.
- Step 7. Evaluation of the putative secondary crater patterns and the possibility of an Evander (or bigger)-scale impactor and its possible effects on Dione's icy surface.

### 2.1. Calculations Supporting the Estimation of the Primary Impactor and Secondary Crater Size

The transient crater size ($D_t$, km) is determined from the observed crater size (D, km) [27,28]:

$$D = 0.7 D_t^{1.13} \tag{1}$$

The size of the primary impactor (d, km) is derived from $D_t$ [25]:

$$D_t = 1.1 \left( \frac{v_i^2}{g} \right)^{0.217} \left( \frac{\rho_i \sin\alpha}{\rho_t} \right)^{0.333} d^{0.783} \tag{2}$$

where d is the impactor diameter in km with a density $\rho_i$ (0.6 g/cm$^3$; [28]), velocity $v_i$ (in the original study—U; for detailed information, please see Section 2.1.1), and incidence angle $\alpha$ (for detailed information, please see Section 2.1.1). $\rho_t$ is the density of the icy crust (910 kg/m$^3$), and g is the acceleration of gravity in km/s$^2$ at the target surface, $2.32 \times 10^{-4}$ km/s$^2$, in the case of Dione.

The average size of the secondary impactors is determined by the following [28]:

$$l_{AVG} = \frac{T}{\rho_t v_e^{2/3} v_i^{4/3}} d \tag{3}$$

where $l_{AVG}$ is the average size (m) of the ejected fragments, and T is the tension fracture, equal to $0.17 \times 10 8$ Pa in ice [29]. As described in the previous equations, d is the size of the impactor (km), $\rho_t$ is the density of the ice at the target location: 0.91 g/cm$^3$ [27], $v_e$ is the speed of the ejecta (for "rubble ejecta": 1.98 km/s; [27]), and $v_i$ is the velocity of the impactor (please see detailed information in Section 2.1.1).

The maximum size of the secondary projectiles was derived from $l_{AVG}$ [28]:

$$l_{MAX} = \frac{m_W + 3}{2} l_{AVG} \tag{4}$$

where $l_{MAX}$ is the maximum size (m) of the ejected fragments, $l_{AVG}$ is the average fragment size (m), and $m_W$ is the so-called Weibull constant, which is 8.7 for ice [29].

The ejected mass ($M_{tot}$; kg) in the case of the theoretical impact (if a secondary crater ring or any pattern is found, and the original impactor/impact crater size can be estimated) is calculated from:

$$M_{tot} = 3.75 \times 10^{-2} \, \rho_t D_t^3 \tag{5}$$

where $D_t$ is the transient crater diameter of the putative crater in km, indicated by the secondary craters, and $\rho_t$ is the density of the ice at the target location: 0.91 g/cm$^3$ [27].

### 2.1.1. Scenarios of the Secondary Crater Formation

The size variation of secondary craters was simulated to determine the characteristics of the impact causing hemisphere-scale renewal and size ranges were provided as a set of boundary conditions indicating various sizes of primary impactors. As mentioned above, two variables were used during the simulation: the impactor's speed and the impact angle. A set of impact velocities was used, including 3.93, 5.81, 7.69 km/s, and 20.4 km/s. The former three represent the collision velocity calculated for main asteroid belt collisions [30], and the latter was applied in various studies targeting asteroid impact reconstructions in icy planetary bodies [26,27]. As a second variable, two different impact angles were set: the average, commonly applied at 45° [26,27], and a relatively low 20° incidence angle, which appeared in a particular study, modeling low-angle impacts and their effects on planetary surfaces [31].

### 2.2. The Studied Location and the Crater Distribution Map

The pivot of the study is the region, located westward from the Eurotas and Palatine chasmata (so-called faulted terrain), defined as one of the intermediate cratered terrains (ICTs) [17]. It spreads approximately between latitude 50° and −50° and longitude ca. 300° to 60° (westward) (Figure 1). The source of the map of Dione is based on Cassini—Voyager Global Mosaic 154 m v1 map and can be found in Astropedia—Lunar and Planetary Cartographic Catalog [32–35]. The applied nomenclature follows the recommendation of the Gazetteer of Planetary Nomenclature [36].

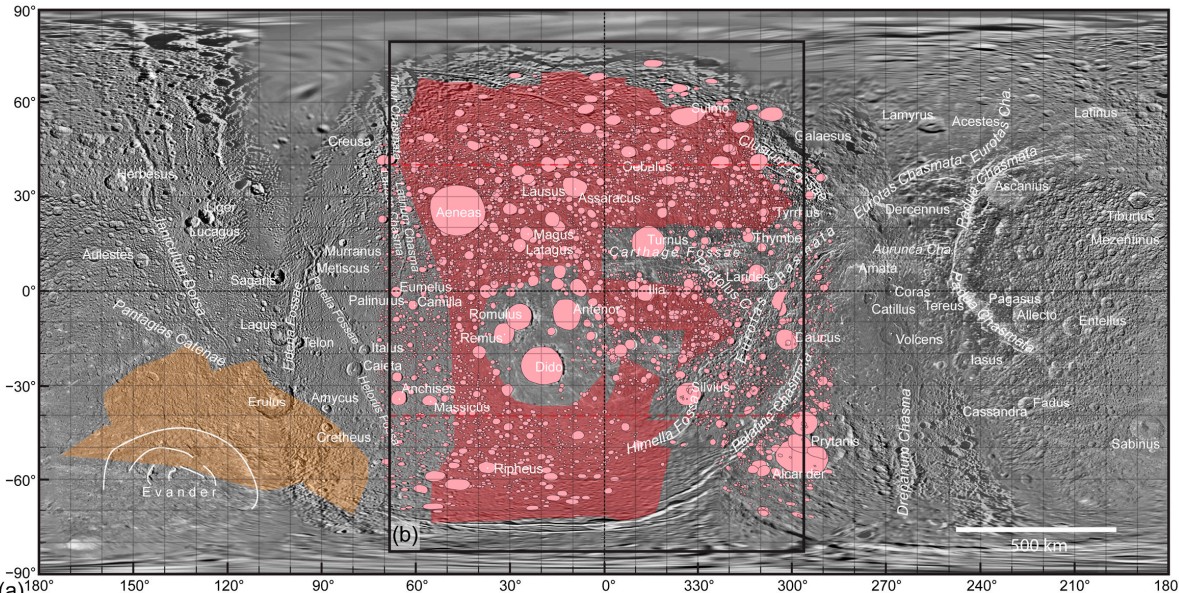

**Figure 1.** (**a**) Cassini–Voyager Global Mosaic 154 m v1 map (can be found at https://astrogeology. usgs.gov/search/map/Dione/Voyager/Dione_Cassini_Voyager_mosaic_global_154m; accessed on 28 February 2023); reprinted/adapted with permission from Refs. [32–35]. Batson, R. 1984; Greely, R. and Batson, R. 2007; Roatsch, T. et al., 2016; and Schneck, P. 2016. The transparent orange polygon indicates Evander Terrain, i.e., the dispersion of the Evander ejectiles [17]; (**b**) the region of study, consisting of the ICT (red, transparent polygon) [17]. Pink ellipsoids indicate the ≥4 km craters used in the analysis. Dashed red lines: areas with crater mapping uncertainties (see Section 2.2).

The remote sensing and GIS research were performed by QGIS 3.22 software ("Firenze" version; released: 21 October 2022; international developer team; https://www.qgis.org/en/site/; accessed on 28 February 2023). In previous studies, the mapping of craters

on Dione was limited to craters with ⌀ ≥ 4 km crater diameter due to the difficulties involved in identifying smaller craters in image mosaics with lower resolution [17]. This study also applies such a criterion to avoid increasing bias in the results due to the use of barely identifiable smaller craters. Over 8800 craters fulfilling the minimum ⌀ ≥ 4 km size limit were identified, including about 5400 craters in the studied area (Figure 1b). The crater distribution maps were created using the craters' centroids and a grid with a 100 km × 100 km basic cell size overlapping the map. The distribution map was based on the number of certain-sized craters appearing in the cells of the grid (Figure 2).

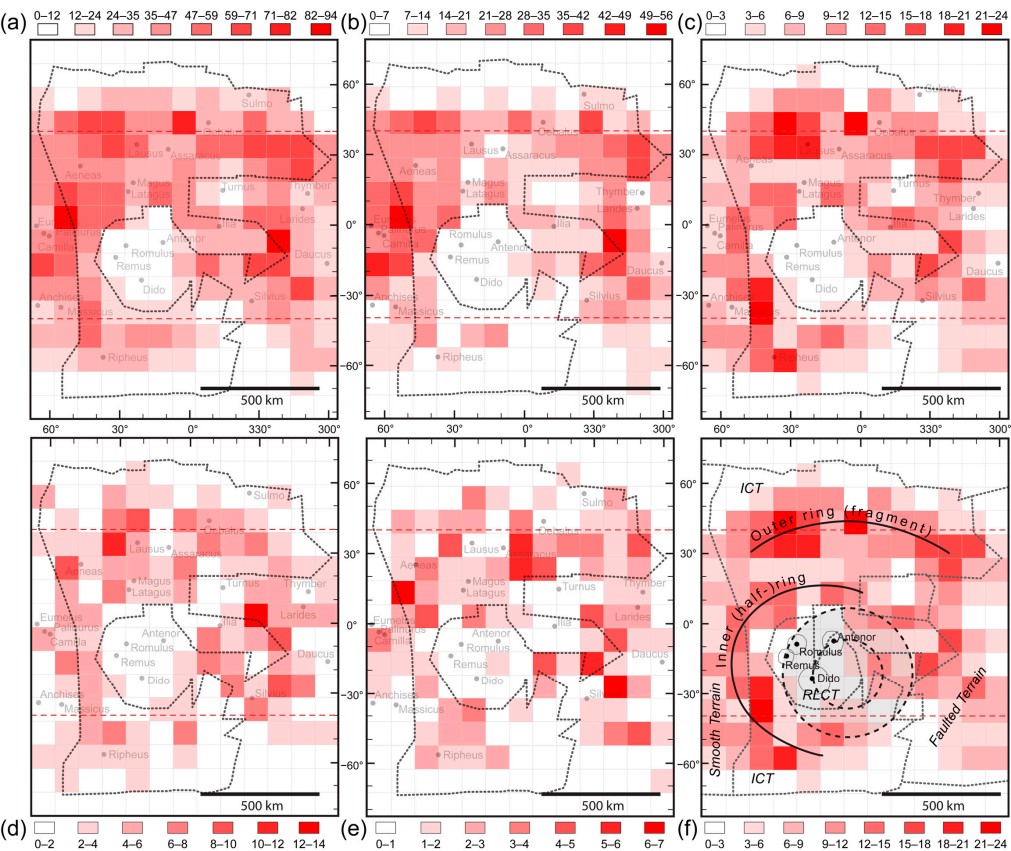

**Figure 2.** The crater distribution maps at the studied intermediate cratered terrain. The distribution maps are based on the map of the identified impact craters introduced in Figure 1b (the pink color polygons mark craters). Crater distribution map of (**a**) all ⌀ ≥ 4 km craters; (**b**) ⌀ 4–5.9 km craters; (**c**) ⌀ 6–7.9 km craters; (**d**) ⌀ 8–9.9 km craters; (**e**) ⌀ 10–11.9 km craters. The dashed red lines: crater mapping uncertainties. The dotted black line (**a–e**): margin of ICT and other terrains (**f**). The cell size of the light grey grid is 100 × 100 km in all figures (**a–f**). Abbreviations: ICT—intermediate cratered terrain, RLCT—recent large-cratered terrain. The grey area with black dashed margins indicates the location of the putative impact crater (inner circle—⌀ 300 km; outer circle—⌀ 550 km). The concentric allocation of craters (ring-like-fragment pattern) may indicate the impact of ejectiles followed by a collision with a giant-size primary projectile (**f**).

Please note that the crater mapping had some limitations in certain areas located between latitude 40° to 90°, and especially between −40° to −90°, due to the quality of the image mosaics and the distortion of the map. This limitation may bias the results, and therefore the areas with such potential uncertainty are marked by dashed red lines in Figure 1b and the maps of Figure 2.

## 3. Results

### 3.1. Simulation of Secondary Crater Formation

Comparing the eight scenarios based on various impact and collision velocities and angles (Scen. 1 to 8; Table 1), the main results can be summarized as followings:

**Table 1.** Relationship between the primary impactor and secondary crater formation on Dione. Columns 1 to 8 show various scenarios using various combinations of impact velocity (U or vi) and angles (ψ), such as 1 (vi: 3.93 km/s; ψ: 20°), 2 (vi: 5.81 km/s; ψ: 20°), 3 (vi: 7.69 km/s; ψ: 20°), 4 (vi: 20.4 km/s; ψ: 20°), 5 (vi: 3.93 km/s; ψ: 45°), 6 (vi: 5.81 km/s; ψ: 45°), 7 (vi: 7.69 km/s; ψ: 45°), and 8 (vi: 20.4 km/s; ψ: 45°). Bold data: primary impactor size; red data: primary impacts which result in ∅ < 4 km secondary craters; grey background: collision scenarios that might have happened in the early history of Dione (based on the crater distribution results, see Section 3.2). The estimated parameters are calculated using Equations (1)–(5) (Section 2.1).

| Crater Size [km] | Simulations (Impactor Size [km] | Secondary Crater Size [km]) | | | | | | | | | | | | | | |
|---|---|---|---|---|---|---|---|---|---|---|---|---|---|---|---|
| | Scen. 1 | | Scen. 2 | | Scen. 3 | | Scen. 4 | | Scen. 5 | | Scen. 6 | | Scen. 7 | | Scen. 8 | |
| 62 (Remus) | **8.0** | 0.9 | **6.5** | 0.5 | **5.5** | 0.4 | **3.2** | 0.1 | **8.3** | 0.9 | **6.7** | 0.5 | **5.7** | 0.4 | **3.3** | 0.1 |
| 81 (Antenor) | **10.9** | 1.3 | **8.8** | 0.7 | **7.5** | 0.5 | **4.4** | 0.1 | **11.2** | 1.3 | **9.0** | 0.7 | **7.7** | 0.5 | **4.5** | 0.1 |
| 90.1 (Romulus) | **12.3** | 1.4 | **9.9** | 0.8 | **8.5** | 0.5 | **4.9** | 0.1 | **12.6** | 1.4 | **10.2** | 0.8 | **8.7** | 0.5 | **5.1** | 0.1 |
| 100 | **13.8** | 1.6 | **11.1** | 0.9 | **9.5** | 0.6 | **5.5** | 0.1 | **14.2** | 1.6 | **11.4** | 0.9 | **9.8** | 0.6 | **5.7** | 0.1 |
| 122 (Dido) | **17.3** | 2.0 | **13.9** | 1.1 | **11.9** | 0.8 | **6.9** | 0.2 | **17.8** | 2.0 | **14.3** | 1.1 | **12.3** | 0.8 | **7.1** | 0.2 |
| 150 | **21.8** | 2.9 | **17.6** | 1.6 | **15.0** | 1.1 | **8.8** | 0.3 | **22.5** | 2.9 | **18.1** | 1.6 | **15.5** | 1.1 | **9.0** | 0.3 |
| 200 | **30.2** | 4.0 | **24.3** | 2.3 | **20.8** | 1.5 | **12.1** | 0.4 | **31.1** | 4.0 | **25.0** | 2.3 | **21.4** | 1.5 | **12.5** | 0.4 |
| 250 | **38.9** | 5.2 | **31.3** | 2.9 | **26.8** | 1.9 | **15.6** | 0.5 | **40.0** | 5.2 | **32.2** | 2.9 | **27.6** | 2.0 | **16.1** | 0.5 |
| 300 | **47.7** | 6.3 | **38.4** | 3.6 | **32.9** | 2.4 | **19.2** | 0.6 | **49.2** | 6.4 | **39.6** | 3.6 | **33.9** | 2.4 | **19.7** | 0.6 |
| 350 (Evander) | **56.8** | 7.5 | **45.8** | 4.3 | **39.2** | 2.8 | **22.8** | 0.7 | **58.6** | 7.6 | **47.1** | 4.3 | **40.4** | 2.9 | **23.5** | 0.7 |
| 400 | **66.1** | 8.8 | **53.2** | 5.0 | **45.6** | 3.3 | **26.5** | 0.8 | **68.1** | 8.8 | **54.8** | 5.0 | **46.9** | 3.3 | **27.3** | 0.8 |
| 450 | **75.5** | 10.0 | **60.8** | 5.7 | **52.0** | 3.8 | **30.3** | 0.9 | **77.8** | 10.1 | **62.6** | 5.7 | **53.6** | 3.8 | **31.2** | 0.9 |
| 500 | **85.0** | 11.3 | **68.5** | 6.4 | **58.6** | 4.3 | **34.1** | 1.0 | **87.6** | 11.4 | **70.5** | 6.4 | **60.4** | 4.3 | **35.2** | 1.0 |
| 550 | **94.7** | 12.6 | **76.3** | 7.1 | **65.3** | 4.7 | **38.0** | 1.1 | **97.6** | 12.7 | **78.5** | 7.2 | **67.3** | 4.8 | **39.2** | 1.2 |

The change in impact angle from the commonly applied 45° [26,27] to a low, 20° incidence angle [31], did not significantly influence the secondary crater size.

The increasing collision velocity significantly decreased the size of ejectiles and, thus, the size of secondary craters. It suggests that, even if a larger impact happened, if the velocity was higher (ca. >10 m/s), the secondary craters would blend into the mass of similar-sized, common primary craters. The secondary crater size would barely reach 1 km, even with a relatively large impactor and high impact velocity (Scen. 4 and 8; Table 1).

The minimum crater size requirement seems to limit the further interpretation of the results, i.e., ∅ < 4 km secondary craters may be formed by large impacts, but they were not mapped due to uncertainty during their identification.

In summary, the crater distribution patterns observed in this study could indicate relatively low velocity, "asteroid-belt-like" collisions between Dione and a minimum ca. ∅ 30–40 km impactor (resulting in a minimum ca. ∅ 200–250 km primary craters).

### 3.2. Distribution Patterns of Various Crater Classes

The distribution pattern of the craters falling in four different diameter-size classes was studied and compared to the general distribution pattern (all ∅ ≥ 4 km craters; Figure 2a).

The ∅ 4–5.9 km crater class pattern is similar to the general distribution, suggesting that even if there is some pattern, it melts into the general crater allocation (Figure 2b).

Two concentric secondary crater allocations (ring-like-fragment patterns) were identified in the distribution map of the ∅ 6–7.9 km crater class (Figure 2c). An inner one at the edge of the relatively crater-clean area, named RLCT (recent large crater terrain), and an outer one, roughly around latitude 30° (Figure 2c,f). The inner concentric secondary crater allocation may result from impacts, forming the larger craters in the area, such as Dido, Romulus, and Remus. The outer concentric ring fragment, if a putative impact created it, maybe the result of a low angle collision (20°) with a ca. ∅ 48–57 km, relatively low-velocity

object (Scen. 1), or a bigger, ⌀ 69–75 km, faster object (Scen. 2). A similar crater pattern may form if a bigger object, ⌀ 49–59 km (Scen. 4) or ⌀ 71–79 km (Scen. 5), collides with the same speeds (3.93 and 5.81 km/s, respectively), but at a greater impact angle (45°) (Table 1).

The distribution patterns of the craters falling into the ⌀ 8–9.9 km and the ⌀ 10–11.9 km crater classes are similar; they consist of some "randomly" located crater groups in the studied area, without any clear pattern (Figure 2d,e).

## 4. Discussion

### 4.1. Surface Renewal Model for the Studied Intermediate Cratered Terrain

Kirchoff and Schenk [17] suggested that one possible explanation for the renewal of the surface in the ICT region was the result of impactors and their ejecta creating the ⌀ ≥ 50 km impact craters. Our working theory suggests that such impactors may be accompanied by a much bigger size, ca. ⌀ 50–80 km, impactor, forming a ⌀ 300–350 to ⌀ 500–550 km crater (Figure 3). Such impactor sizes may have a complex influence on the surface and subsurface region of the satellite, which can be summarized as the following.

- Ejecta blanket. During the excavation phase of such a collision, $3.3–5.5 \times 10^5$ t (⌀ 300–350 km crater) to $1.3–1.7 \times 10^6$ t (⌀ 500–550 km crater) of debris might be ejected into space and return to the surface, covering large areas of the ICT with an ejecta blanket (Figure 3b,c).

- Ricocheting debris. In addition to the ejecta blanket, in the case of a low-angle collision, the sliding and ricocheting ejectiles [31] might cause surface planing by "plowing" and partial melts.

- Intensification of cryotectonic and cryovolcanic activity. Following the excavation phase of the impact, an impactor of such a size may cause the uplift of the ice crust (rebound) and the rise of a subsurface diapir-like structure made by the convective ice layer and/or the cryo-slurry at the center of the impact crater during the modification stage of the impact (Figure 3d,e). The diapir and central peak formed in the convective ice crust might later retreat due to isostatic relaxation of the surface. Diapir formation may cause the intensification of cryotectonic and cryovolcanic activity in the region, accompanied by faulting and cryo-slurry outflows. Such secondary processes might have a significant role in the surface renewal of ICTs (Figure 3d,e). Analog putative impact-induced cryovolcanic activity was hypothesized in the case of Europa, where circular fractures ("spider-like" landform) with central depressions were described as the result of impact-induced brine pocket migration, which results in the concentration of aqueous melt and plume-like cryovolcanic eruptions [37]. Regarding the role of cryotectonic and cryovolcanic activity in surface renewal, it might be limited to smaller areas in the region (neighborhood of the primary impact crater), considering the existence/preservation of the putative secondary impact craters that were most likely not affected by the impact-related endogenic processes.

- The hypothetical connection between younger craters in the region and the putative impact. As shown in Figures 2f and 4b, the supposedly young craters of the so-called recent large-cratered terrain (RLCT) and the location of the putative giant impact overlap. The putative huge impact site seems to be "hidden" under the large impacts found in the RLTC. The allocation of such overlap and the unusually high abundance of large younger impacts in the region raise the question about the possible connection between the putative giant impact, its effect on the ice crust, and the formation of the younger impact craters. Computer simulations of Lunar crater formation showed that in areas where the temperature nearside of the crust and upper mantle is hotter (thinner crust), impacts might form craters up to twice the diameter of the craters formed at the "cooler" side [38]. If such a model applies to the formation of the large RLTC craters, it might suggest a thermal anomaly in the icy crust under RLTC. Such a thermal anomaly may be some residual heat or may appear due to the unusually thin crust, as possibly the effects of the putative giant impact.

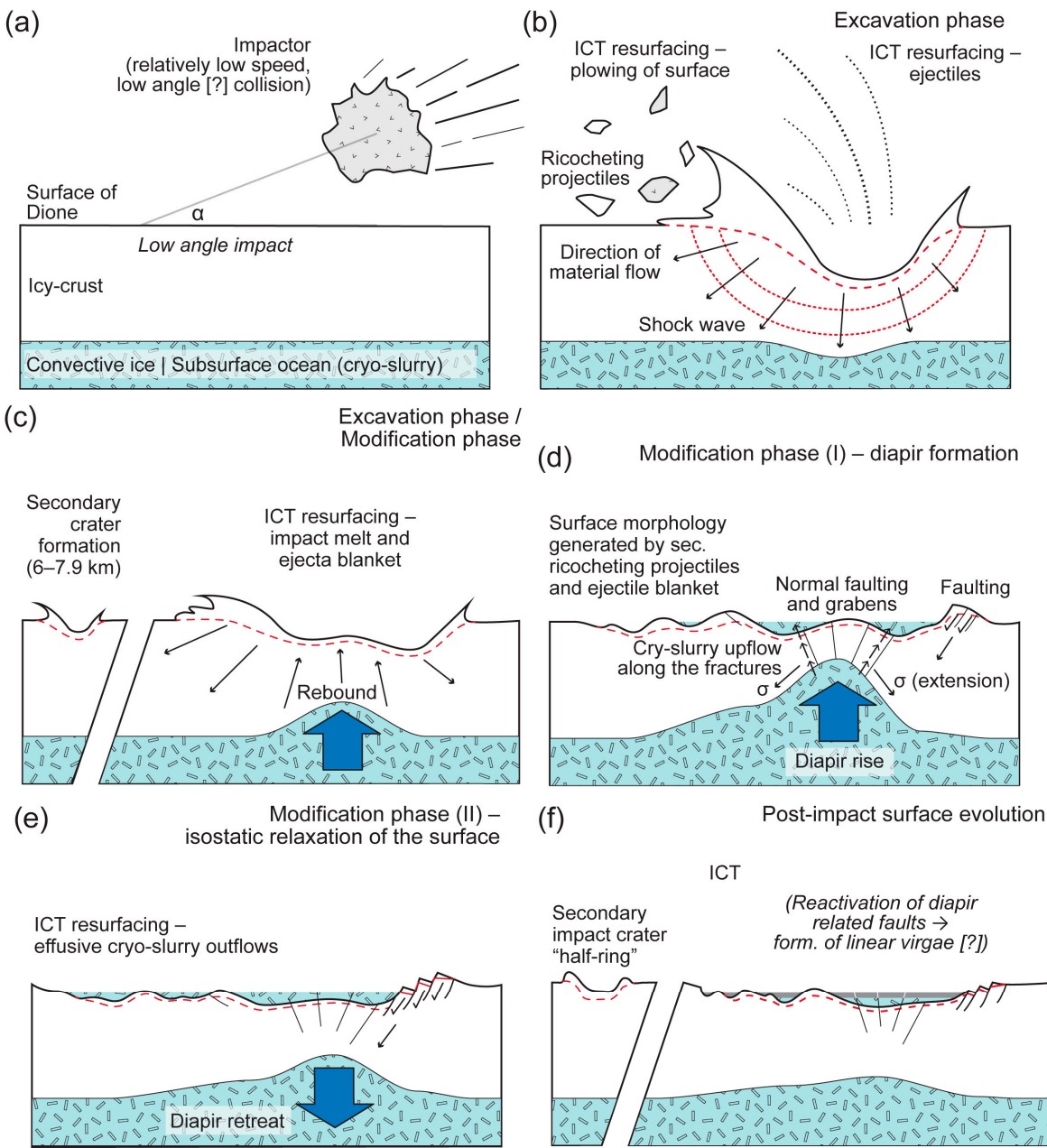

**Figure 3.** The main steps of surface renewal of intermediate cratered terrain following an asteroid impact. (**a**) Low-angle collision; (**b**) excavation phase of the impact and "plowing" of the ICT's surface; (**c**) excavation/modification phase transition and the ongoing deposition of ejecta blanket; (**d,e**) modification phase with diapir formation and intensification of cryotectonic and cryovolcanic activity; and (**f**) postimpact surface evolution. Please note that regarding the existence and preservation of putative secondary craters, the cryotectonic and cryovolcanic processes might be limited around the primary impact crater (Section 4.1). The mark "[?]" indicates the theoretical nature of the statement: (**a**)—the possibility of a low-angle collision; (**f**)—possible link between the formation of linear virgae and post impact cryotectonic processes.

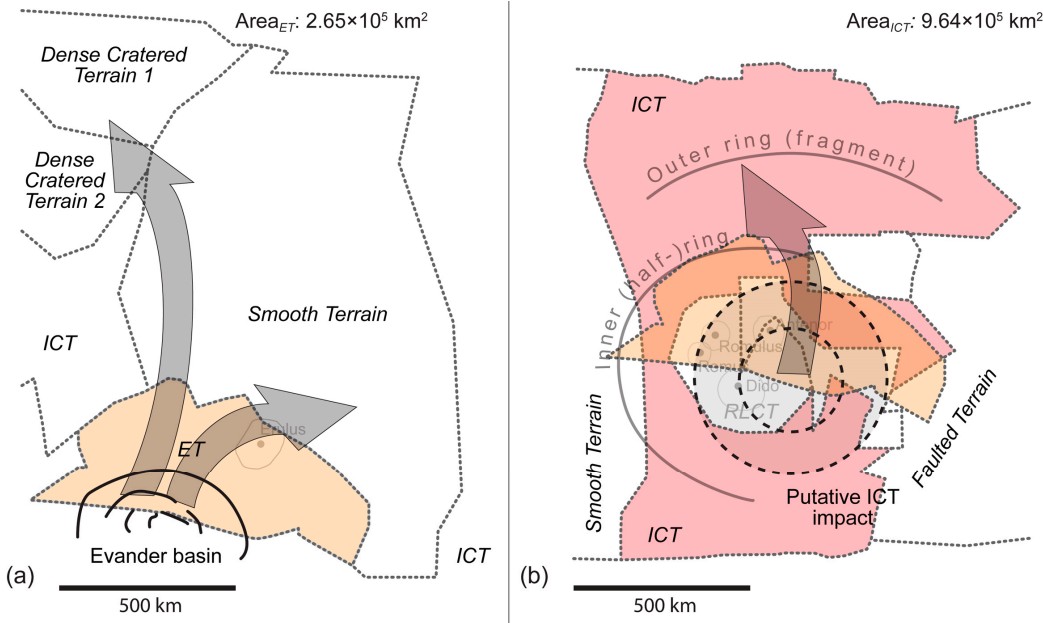

**Figure 4.** The simplified map, explanation, and comparison of the effect of Evander (**a**) and the putative ICT impact (**b**). The transparent orange indicates the area of Evander Terrain, i.e., the dispersion of the ejecta following the Evander collision. The transparent reddish pink area shows one of the intermediate cratered terrains, the studied area. Abbreviations are ET—Evander Terrain, ICT—intermediate cratered terrain, and RLCT—recent large-cratered terrain. The grey arrows indicate the dispersion of secondaries and the location of suspected secondary impact craters.

### 4.2. Some Thoughts about the Origin of the Impactor

Here, we introduce a working theory about a giant impact on the surface of Dione, which might have contributed to the resurfacing of ICT, a region with an unrevealed astrogeological history. There are some possible timing and sources of the impactor, summarized in brief below.

Although Kirchoff and Schenk [17] found that the age of the most prominent craters of Dione (namely Aeneas, Unnamed, Acestes, Erulus, Allecto, and Dido) seemed younger than the outer Solar system's late heavy bombardment (LHB) period (~3.9–4.1 Ga) [39,40], they also mentioned that the calculated age might be younger than the formation age of the craters, regarding, e.g., the degradation of the craters' morphological features [17]. Suppose such craters are older than their calculated age suggests [17]. In that case, it might be possible that the disturbance, triggered by the late migration of the giant planets, caused increasing asteroid activity, not only in the inner but in the outer Solar system as well [39,41–43]. Please note that there has been an ongoing debate about the Late Heavy Bombardment, e.g., due to the lack of observational evidence of a late increase in the lunar bombardment rate [38,44]. For this reason, it may be better just to refer to heavy bombardment, indicating the early history (first billion years) of the planetary system in which the impact rate was significantly higher than during the last 3.5 Ga.

Collisions appearing at the phase of giant impact [45] in the Saturnian system may have fed the group of potential impactors, which may have hit Dione during its early history. The period is dated back to ca. 4 Ga [45] and may also coincide with the (late) heavy bombardment, which would describe its violent nature. The giant impact phase is described as a period when the originally "Galilean-like" satellite system collided and merged, forming Titan and the other icy satellites [45]. Regarding the ICT's 3.5 + 1.0/−2.6 Ga age, it might have been exposed to intense asteroid bombardment, including the putative basin forming impact at the later part of the giant impact phase.

The third possible scenario in the search for the potential source of the impactor points toward a unique Saturnian impactor population [46,47]. Although there are some uncer-

tainties about the active impact period of the asteroids, originating from the suggested Saturn-specific planetocentric debris, elliptical craters represent the collisions that frequently appear on ICT. Therefore, the Saturn-specific planetocentric origin of the identified putative ICT impactor cannot be excluded from the listed scenarios.

Referring to the analogy between the putative impact and Evander (Section 4.3), it is also worth mentioning the following potential source of impactors. Although the putative crater and Evander are located in different latitudes, such an impact might fit a hypothetical capture of a Kuiper-belt object fragment crossing the orbit of Saturn (e.g., small Solar system bodies, known as Centaurs) [48]. Another impactor possibility would be a high-inclination and long-period comet. A disruptive interaction during its travel would form "smaller" fragments in both cases, such as the Centaurs and the comet (such as Shoemaker−Levy 9). It could also reduce the projectile's velocity to match the preferred low-velocity encounter.

### 4.3. The Evander Analogy

The impact resulting in the formation of the Evander basin seems the most fitting candidate to find some analog to the putative impact, which might have caused the surface renewal of the ICTs, including the target region of this research. The Evander basin, a multi-ringed, relaxed impact structure, is in the so-called Evander Terrain [17]. The formation of the Evander Terrain, i.e., the time of the impact, is dated back to <1 Ga (likely <2.5 Ga), and the relaxed character of the crater/basin suggests that Dione was thermally active from that period until very recently, but at least half of its recorded history [17]. The possible analogies between the Evander impact and our putative impact can be summarized as follows.

Physical parameters (size) analogy. Based on the modeled dimensions of the putative ICT impact, due to its approximate 350 km diameter size, the Evander basin falls into the smallest-size impact category (300–350 km diameter), which might be responsible for the ICT surface renewal (Table 1). For such reasons, an Evander impact may be a good candidate as an analog process.

Ejecta blanket analogy. The study of the surroundings of Evander's ejecta blanket indicated that although the impact covers the Evander Terrain and its neighborhood, the larger craters appearing at the edge of the blanket are not fully covered by Evander's debris [17]. Based on the size of the Evander basin, the impact draped the surrounding area in approximately $3.3–5.5 \times 10^5$ t material, which, based on the size of the Evander Terrain [17], covered an area of about at least 30% of the size of the studied ICT, located around the center of the putative impact identified in this study (Figure 4). Based on the results of this study, the maximum amount of ejectiles of the putative impact might have reached $1.3–1.7 \times 10^6$ t, which most likely covered the entire ITC, contributing to the renewal of the surface (Figure 4). (Please note that due to some possible crater mapping limitations—resolution/distortion problems—the area of ET may not fully represent the maximum dispersion of the ejectiles in [17]). It may be an odd coincidence, but worth mentioning that if the dispersion area of the Evander ejectiles (i.e., the Evander Terrain) is used to represent a dispersion area for a same-size putative impact at the ITC, the edge of the dispersion field and the identified inner crater ring overlaps (Figure 4). As an analog of the Evander ejecta blanket, the identified inner crater ring may indicate some boundary between the area intensively affected by the putative impact's ejecta and the region located further from the center of collision (Figure 4b). In addition, in the case of the ICT, maybe only the smaller-size craters were fully covered by the ejecta blanket. Still, to verify this theory, further surface examinations are needed.

Analogy with the fate of the secondaries. During the explanation of crater size-frequency distribution on various terrains, Kirchoff and Schenk [17] mentioned the role of the impacts of Evander's secondaries on various terrains, namely dense cratered terrain 1 and the neighboring smooth terrain. The former is located at the opposite pole of the satellite, which suggests a longer distance secondary bombardment, along with the also suggested self-secondaries, which would fall on the area of the Evander Terrain itself [17]. Such results, although there are still some unanswered questions (e.g., the reason why

no influence from Evander's secondaries was recognized in closer terrains, such as the studied ICT), suggest the satellite-scale dispersion of the secondary impacts in the case of an Evander-size (or even bigger) collision, and its possible role in the surface renewal of certain areas.

Analogy with crater morphology I—multi-ringed structure. Among numerous theories that aim to explain the formation of multi-ringed crater structures, there are some which, along with the postimpact viscous relaxation processes (see below), may appear in the case of Evander and the putative ICT impact as well; even such structures cannot be recognized in the case of the latter. Two alternatives of multi-ringed crater formation may be considered in the case of Evander, namely, (i) the modification of the original crater shape by the volcanic intrusion and (ii) the impact on the layered target, followed by layer-specific collapse processes and the formation of concentric craters [49]. In the case of Dione, the former may appear as viscous, convective ice, or as a cryo-slurry diapir. Along with the cryovolcanic intrusion scenario, the latter may support the existence of two different ice layers in the ice crust, mentioned initially by Zhang and Nimmo [9], or provide evidence for a subsurface ocean. The same applies to the third so-called tectonic theory, which describes the multi-ring formation as a result of the inward flow of an underlying material (mantle) during the collapse of the transient cavity formed during the collision. Such inward flow pulls the overlaying unit (crust) with it, causing the formation of ring-aligned extensional (normal) fault systems [50,51].

Analogy with crater morphology II—viscous relaxation. Viscous relaxation of the surface as a process appearing on icy satellites has been recognized since the 1980s [46]. Viscous relaxation results in low relief on the surface of icy satellites, and shallow craters, often with domed floors and sharp, well-defined rims [52]. Such morphology may appear on icy satellites with an ice crust characterized by uniform viscosity or viscosity, which decreases with depth. Viscous relaxation works differently in an ice crust that is thin compared to the crater diameter or in the case of thick crusts, where the viscosity increases with depth [46]. Independent of the possible scenarios, the morphological mark of viscous relaxation provides information about the thermal history of the satellites. Dione is not exceptional; the relaxed crater morphology of Evander [53–55] suggests that the satellites were thermally active during the formation of Evander (<1 Ga, or <2.5 Ga) and most likely during the putative ICT impact, which, based on the age of the terrain, dated back to around 3–4 Ga [17]. Given the maximum >500–550 km diameter size of the impact crater (Table 1), the impact crater (or its characteristic morphologic features) might disappear relatively "fast" from the surface due to the viscous relaxation of the icy crust [52].

## 5. Conclusions—Summary and Final Remarks

The study discussed the possibility that a minimum Evander-size impactor, which could be responsible for the complex resurfacing processes, might appear in one of the intermediate cratered terrains of Dione.

A working theory is provided concerning the fate of the impact from the collision until the possible disappearance of the characteristic morphologic features of the impact basin due to the viscous relaxation of the ice crust, most likely accompanied by further, smaller sized impact cratering.

There are three potential timings/sources of the impact, including asteroid activity during the (late) heavy bombardment period and the so-called giant impact phase, a period during the early evolution of the Saturnian system. Along with the high probability of collisions in the mentioned periods, a unique, planetocentric-impactor population in the Saturnian system may be counted as a potential source of the impactor.

Despite the assumption concerning the surface renewal of the ICT based on the analysis of its crater age and the crater distribution, which may indicate a characteristic secondary crater pattern, there is not much direct evidence that supports the theory for a putative ICT impact. The similarities in simple physical parameters (i.e., size, the characteristics of the dispersion of the ejecta blanket and secondary impactors, and the possible secondary effect

of a basin-scale impact, such as cryotectonic activity and viscous relaxation) simply do not help to explain the fate of the putative ICT impact but may provide information about the inner structure of the satellite, including the possible units of the ice crust and the existence of the subsurface ocean. For such reasons, Evander as an analog plays a vital role in this and further studies.

Future research aims to complete more simulations of the effect of a giant impactor on the surface of Dione, a possible astrogeological mapping and a detailed comparison of the surroundings of the putative impact and the Evander crater, Dione's largest impact crater, with a similar size to the estimated size range of our theoretical crater.

**Author Contributions:** Conceptualization, B.B.; formal analysis, B.B., M.N. and C.G.; investigation, B.B. and M.N.; methodology, B.B. and C.G.; supervision, B.B.; visualization, B.B.; writing—original draft, B.B. and C.G.; writing—review and editing, B.B. All authors have read and agreed to the published version of the manuscript.

**Funding:** This research received no external funding.

**Data Availability Statement:** Data will be available upon request.

**Acknowledgments:** We would like to thank the reviewers of this article for taking the time and effort to review the manuscript. We appreciate all their valuable comments and suggestions, which helped to improve the quality of this manuscript.

**Conflicts of Interest:** The authors declare no conflict of interest.

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
