# Peer review of "A Theory about a Hidden Evander-Size Impact and the Renewal of the Intermediate Cratered Terrain on Dioneâ€"

_universe, doi:10.3390/universe9060247_

Round 1
Reviewer 1 Report
This manuscript proposes a crater excavation theory to explain the origin of the Evander-size impact on the surface of Dione. I found the manuscript of high quality and quite interesting for a general audience.
A quite important point to improve. In the introduction you introduced a simplistic description of the formation of Jupiter and Saturn satellites as given in lines 39 and 40: " Without going into details, icy satellites most likely formed out of the “leftover material” of gas (Jupiter and Saturn) and ice (Uranus and Neptune". Today we know that both giant planets developed disks of materials in a similar way than the Sun formed the planets (see e.g. Makalkin & Dorofeeva, 2014 and references therein). The first building blocks of Jupiter and Saturn satellites were probably C- and water-rich chondritic planetesimals. The environments surrounding giant planets could also be source of solar-nebula instabilities where water-rich comets could accrete as well (Blum et al., 2016).
In reference to the formation of the Evander crater at low latitudes of Dione, such an impact might fit a hypothetic capture of a Kuiper Belt Object fragment crossing the orbit of Saturn, objects known as Centaurs (see e.g. Jewitt, 2008). Other possibility is a high inclination long period comet. In both cases, a disruptive previous interaction like the one occurred with Shoemaker-Levy 9 could reduce the velocity of the projectile to match the preferred low-velocity encounter.
In reference to the formation of craters and the depth of the final regolith layer, the authors might be interested in the conclusions obtained by Beitz et al. (2016) about the crater size as a function of the projectile size. Significant regions of Dione might be buried by the ejecta blankets of the largest impacts, explaining some of your observations about the terrains surrounding Evander.
In general I found the manuscript quite attractive, example of good scientific skills, and with enough quality to deserve publication in Universe.
Additional suggested references to be included:
Beitz E. et al. (2016) "The collisional evolution of undifferentiated asteroids and the formation of chondritic meteoroids", Astrophysical Journal 824, art.id.12, 29 pp.
Blum J. et al. (2014) "Comets formed in solar-nebula instabilities! - An experimental and modeling attempt to relate the activity of comets to their formation process", Icarus 235, 156-169.
Jewitt D. (2008) In Transneptunian Objects and Comets. Springer-Verlag, Germany.
Makalkin A.B. & V. A. Dorofeeva (2014) "Accretion disks around Jupiter and Saturn at the stage of regular satellite formation", Solar System Research volume 48, 62–78.
Reviewer 2 Report
Review of the manuscript: A theory about a hidden Evander-size impact and the renewal of the Intermediate Cratered Terrain on Dione submitted to Universe by Bradak et al.
The manuscript reports on a theory to explain surface features of the icy satellite Dione, orbiting Saturn. The crater distribution on Dione is shown as a global map (Fig 1) and maps showing the abundance of craters with different sizes (Fig. 2).
In addition, the sizes of secondary craters in impact evets of different sizes and impact angles are calculated and tabulated (Table 1).
The manuscript reports on an interesting idea, that on icy satellites the record of very large craters could have been erased due to the post impact modification processes and crustal renewal. And that these processes might operate more efficiently on icy bodies compared to the rocky surfaces of the terrestrial planets. This sound reasonable and the described scenario can serve as concept worthwhile to consider.
The manuscript is well written. Nevertheless I found it difficult to understand and had to read a few times during writing the review. I think the clarity of the manuscript should be improved. I am not sure how much the calculated sizes of the secondary craters serve as a proof of concept for the presented idea. It seems more like a set of boundary conditions for the described event, and a first test of plausibility. I don’t have the expertise to evaluate the mathematically correctness of the calculations and assume that they are OK.
Overall I think the manuscript can be a good contribution to understand the geological history of the satellite Dione orbiting Saturn.
Some general comments are provided below.
Material and Methods:
Did you perform numerical models and if so with which program? Or are the results n Table 2 calculated using the equations 1 to 5 given in method section?
Figure 2: Is this the observed crater population on Dione. I first thought that this was a model result of an expected distribution of secondary crater. I had difficulties to understand what is shown in figure f and additional explanations should be added in the figure caption (not only in the text). Why is the grid size in part of figure f stated separately?
In the results it is discussed a ring pattern related to the abundance of 6-7,9 km diameter craters. Why should secondary craters from a ring like pattern? Are there any examples from other planetary bodies showing a preferentially ring like pattern of secondary craters? I am not aware of any such findings.
I did not work or read much about the geological history of Dione. But looking at the image in figure 1 it appears to me that the surface is older than other regions because it shows more craters. The terrain displays a high abundances of both big and smaller craters compared to other regions of Dione, which looks like a typical size frequency distribution of impactors. I think the authors should better explain why the crater distribution should be unusual and not simply the result of the age of the surface. For example, why should the impact record of the ICT be dominated by secondary’s also the size frequency distribution is not unusual, besides a lower number in <15 km diameter craters compared to DCT1 (see Krirschoff Schenk 2015 Icarus). Would you expect a specific signature in the size frequency distribution of secondary craters compared to the primary impact flux?
How should the impact event renew the ICT surface area and at the same time produce an intense secondary crater field? Does the surface shows indications for a massive ejecta blanket in that region or features similar to those surrounding the Evander basin (in case there are any)? Later processes such as during the crater modification phase or due to initiated “magmatic” processes as indicated in fig. 3 would affect both the previous primary and the secondary crater population to the same degree.
Alternatively it might be that the region called RLTC is the fully renewed area and that the ICT is older. In your scenario it might be that only the smaller sized craters were obliterated by the ejecta blancket of the proposed very large crater.
I assume that in your model the RLTC is proposed to be the very large crater. As I said, I am not familiar with the geological evolution of Dione. But reading shortly the Krirschoff Schenk 2015 Icarus paper it seems that this RLTC region is also a bit unusual as it hosts four recent large craters; an unusual high number. My first thought is that this region could have specific crustal properties (such as a thin crust). Meaning projectiles of a given size would make a significantly larger crater in the RLTC compared to other region of Dione, hence explaining the surprisingly high concentration of the large craters (Antenor, Dido, Romulus, and Remus ) in that are area, an area which look rather young. In this context you may like to check this study regarding the uneven distribution of lunar basins, mainly located on the equatorial lunar nearside (https://www.science.org/doi/10.1126/science.1243224).
For a massive impact event onto a satellite like Dione there might also be secondary craters due to ejecta that first goes into orbit around Saturn before re-impacting onto Dione. Are such late secondary craters part of your model or are they omitted, maybe because of having a random distribution with some focus on either the leading or tailing side of the satellite. Does your models give some information for the larges ejecta accelerated into a Saturn centric orbit? If so it would be interesting to add such information.
In the paper the author often mention the Late Heavy Bombardment. This concept might still be preferred by some authors but to me this concept seems outdated. There is no observational evidence for a late increase in the lunar bombardment rate (see abundance of “old lunar impact ages (Fernandes et al. 2013 DOI: 10.1111/maps.12054) and the work by K. Miljković et al. 2013 https://www.science.org/doi/10.1126/science.1243224 shows that that the projectile sizes for the largest lunar basins seems to be overestimated. Overall the often cited work by Bottke, Morbidelli and coworkers seems to be unnecessarily complicated. With complicated I mean the very demanding (low probability) assumption that the Solar System remained in a meatstable state for 400 million years. Especially considering the lack of evidence for a late increase in the lunar bombardment rate I think it is better to talk about a Heavy Bombardment meaning the early history (first billion years) of the planetary system in which the impact rate was significantly higher than during the last 3.5 Ga.
Round 2
Reviewer 2 Report
Review of the revised manuscript: A theory about a hidden Evander-size impact and the renewal of the Intermediate Cratered Terrain on Dione submitted to Universe by Bradak et al.
The revision addressed my previous comments to an acceptable degree. The figure caption 2 is improved in clarity and the aims of the study are now presented in a way making it easier to understand.
Still I don’t know why secondary ejecta should produce a ring like pattern around the putative crater.
But overall the manuscript is suitable for publication in the journal Universe.
